# Relationship between demoralization of the college student with their individual- and social-oriented self

Chuan-Yung Huang[1], Shun-Hao Hu[1], Li-Ying Fan[2]*

1 Department of Thanatology and Health Counseling, National Taipei University of Nursing and Health Sciences, Taipei, Taiwan, 2 Department of Education, National Taipei University of Education, Taipei, Taiwan

* fly@mail.ntue.edu.tw

**Data Availability Statement:** All relevant data are within the paper and its Supporting information files.

## Abstract

Demoralization has become increasingly prevalent among college students who have lost motivation in life and feel hopeless about their future. Many college students who demonstrate symptoms of demoralization are neglected because they might fail to typical symptoms of depression. Taiwanese college students are simultaneously influenced by bicultural-self system, such as individual- and social-oriented views of self, which vary considerably in the view of self, achievement motivation, and the value of self-realization, and may even create contradictory expectations and behavioral standard. The purpose of this study was to investigate the extent of the relationship between attitude towards demoralization, individual- and social-oriented views of self. Three-hundred fifty-six college students completed the online questionnaire, which was designed to explore their demoralization status and cultural differences. Four groups were divided into bicultural self, individual-oriented self, social-oriented self, and unintegrated self. Bicultural group demonstrated significantly lower demoralization overall scores than other groups. Moreover, the five dimensions of demoralization in college students were mostly significantly and negatively correlated with individual- and social-oriented views of self, indicating that college students' bicultural views of self may contribute to or prevent demoralization.

## Introduction

In college, many students seem to be confused regarding their future, career, life purpose, or even feel that no goal is worth their efforts. However, they do not meet the diagnostic criteria for depression. Demoralization is more accurately to reflects the condition in the college students who have lost morale and hope. It is noteworthy that demoralization is conceptually different from depression.

Demoralization and depression are two different entities whereby the former involves helplessness and hopelessness and the latter entails anhedonia [1–3]. Moreover, depressed people are hardly joyful during the present and future times whereas demoralized people could be

**Funding:** The authors received no specific funding for this work.

**Competing interests:** All authors have declared that no competing interests exist.

joyful at the present moment but feeling gloomy towards the future [4]. Demoralization can be regarded as the result of individuals' difficulty in developing meaning in life [5], referring to a goal that gives individuals a sense of direction and value. Therefore, the social environment in which people live determines the type of people they wish to become [6, 7], and the social expectations and the cultural values might influence people's view of self and the meaning of life [8].

Cultural values directly affect the development of the meaning of life in college students, and two substantially different cultural value systems coexist in Taiwan, namely social-oriented and individual-oriented cultural values [9]. The individual- and social-oriented views of self can coexist in a person. Yang [10] suggested that the values of submission to authority, conservatism and endurance in Taiwanese college students. Male dominance has been gradually replaced by gender equality, but filial piety and ancestral worship as well as fatalism and defensiveness still coexist with optimism and assertiveness, independence and self-reliance, and respect for emotions despite changes in the social environment. In addition, previous studies have reported that self-construal in Taiwanese people is neither completely interdependent (traditional values) nor completely independent (modern and Western values); instead, Taiwanese people develop a composite self in which the two cultural value coexist. Each of the two cultural value systems can form the basis of an individual's ideal self and thereby provide meaning in life, but whether the two can be effectively integrated remains uncertain. As college students enter the workforce where two contrasting cultural value systems coexist, whether they manage to develop a composite self or bicultural self that integrates two systems is of great importance [9]. The coexistence of two value systems may lead to intergenerational cultural and value conflicts within college students. However, the causes and countermeasures of the currently prevalent demoralization among Taiwanese college students have yet to be confirmed by relevant research.

The present study thus aimed to investigate how college students' demoralization influence by cultural values. In the following sections, we first briefly reviewed the definition of demoralization followed by the review of the existing literatures of demoralization and social-/individual-oriented cultural values. Then, we provided an overview of the present study.

## Demoralization

Demoralization has been described as a psychological state characterized by helplessness, hopelessness, a sense of failure and the inability to cope (for a review, see [11]). Demoralization is defined as troubles in lives whereby affected individuals would feel helpless, meaningless, hopeless, remorse and so on when facing life problems [1, 2]. Clinically, demoralization is a new diagnostic concept when they discovered that patients with affective disorders and patients with schizophrenia showed the same psychological condition: loss of meaning, goals, and hope in life; a pessimistic attitude toward the future; alienation and lack of support; and lack of motivation to navigate through life [12]. Previous studies have purposed a various psychological well-being dimensions which are associated with an individual's development of optimal functioning: positive evaluation of one's self, the belief that life is purposeful and meaningful, the possession of quality relationships with others, a sense of continued growth and development, a sense of mastery over one's environment, and a sense of self-determination [11, 13]. However, demoralization is opposed to psychological well-being. Previous studies have showed that patients demonstrated worse psychological well-being scores and significantly higher rates of demoralization [11, 14, 15].

Kissane et al. [12] suggested that demoralization is a manifestation of existential distress, and they developed the first version of the Demoralization Scale, including five dimensions:

such as loss of meaning, despair, disheartened feelings, helpless feelings, and a sense of failure. The scale was consisting of 24 items with 5-point Likert scale for participants' self-report demoralization syndrome. In their study, they recruited 100 patients with advanced cancer (47 males and 53 females) to examine the internal consistency (Cronbach's alpha coefficients ranged from .71 to .94) [16]. Moreover, factor analysis revealed five factors: loss of meaning (loss of meaning and worth connected to role and value in life), dysphoria (unspecific emotions of grief and regret), disheartenment (sense of isolation connected to frustration and loneliness), helplessness (subjective incompetence involving loss of control, hope and so on) and sense of failure (reversed sense of accomplishment, satisfaction and success in life translating into sense of failure).

Hung et al. [2] translated Kissane's Demoralization Scale into Mandarin version and tested its reliability and validity. They used the same five factors as Kissane et al. [16], and the .928 Cronbach's alpha was reliable. Hung [17] modified the Mandarin version of the Demoralization Scale for investigating teenagers self-report demoralization syndrome, and five factors were meaning of life (satisfaction with the current life and self-identification), loneliness and helplessness (subjective feelings of isolation, lack of support or inability to receive assistance from others), self-assurance (sense of self-efficacy and understanding one's own capability and strengths), bravery and perseverance (willingness to face difficulties and shoulder responsibilities), and emotional distress (feelings of regret, dysphoria, and lacking control).

Hung [17] revised the scale to measure demoralization among high school students and to widen the application of the concept of demoralization. The demoralization experienced by high school students is similar to college students. Demoralized high school students and college students both experience subjective depression and agony, feel that no one understands them, and exhibit a certain level of suicidal ideation. However, some college students might show symptoms of depression such as loss of interest and social withdrawal. The differences between demoralization and depression inspired us to explore and gain a greater understanding of demoralization. Despite patients' demoralization has been well documented, knowledge about normal population and the related factors is still evolving. Hence, investigating college students offers a better understanding of their demoralization status and the cultural influences.

## Individual- and social-oriented views of self

Meaning of life is influenced by social expectations derived from cultural values, which shape individuals' view of self [5, 7, 8]. Cultural values directly affect the development of meaning of life in college student, which in turn affects their demoralization level. In the present social environment of Taiwan, two substantially different cultural value systems coexist, namely social-oriented and individual-oriented cultural values [9]. We postulated that the coexistence of two cultural value systems causes cultural value conflicts in college students and contributes to the development of demoralization. Yang [18] defined social orientation as the combination of high fusion and low self-determination and individual (personal) orientation as the combination of high self-determination and low fusion. The interaction between Chinese people and their living environment is mainly social-oriented, whereas the interaction between Westerners (especially Americans) and their living environment is mainly individual-oriented. After compiling the aforementioned discourse on the individual- and social-oriented views of self, Lu [19] argued that the two are different in ontology and structure, function and operation, and self–society relationship.

The ontology and structure of the individual-oriented view of self is characterized by treating the self as a homeostatic and tangible entity, emphasizing individual uniqueness, valuing

differences, maintaining consistency between the public self and private self, and considering the pursuit of self-transcendence as the ultimate expression of humanity. The ontology and structure of the social-oriented view of self is characterized by treating the self as a changeable and flexible entity, emphasizing the similarity between people, valuing the ordinary, and accepting—even encouraging—the difference between the public self and private self, pursuing self-expansion to include others, and achieving the ultimate goal of unity between nature and man.

In the individual-oriented view of self, social exchanges are regarded as the foundation of social relationships, emphasizing interpersonal competition, stressing the importance of rationality, and valuing expression of personal preferences. In the social-oriented view of self, symbiosis is regarded as the foundation of social relationships, emphasizing shared interpersonal relationship, stressing the importance of interpersonal affection and the practice of appropriate behavior and face-saving skills, and being sensitive to other people's responses and evaluation, which comprise the social information from which self-knowledge is formed.

For the self–group relationships, the individual-oriented view of self emphasizes self-reliance, deems joining a group as a personal choice, stresses the importance of maintaining in-group psychological distance, prioritizes the pursuit of personal goals and pleasure, and emphasizes personal well-being. By contrast, the social-oriented view of self centers on belonging to and integrating into social groups, stresses the importance of in-group integration, and prioritizes the pursuit of group goals and the well-being of the group even at the expense of personal interests.

For the self–society relationship, the individual-oriented view of self advocates the antinomic relationship between self and society, considers personal emotions and attitudes as a behavioral guide, emphasizes dominance over the environment, and relies on self-reinforcement and self-growth to gain self-esteem and social respect. By contrast, the social-oriented view of self advocates the integrated relationship between self and society, abides by social rules and norms, emphasizes sensitivity to situations, demands that individuals assume social roles and fulfill the responsibilities of such roles, emphasizes adapting to the environment, and relies on self-criticism and commitment to self-improvement to avoid social punishment and gain social affirmation.

Lu [19] systematically distinguished the differences between these two cultural values (i.e., individual- and social-oriented views of self) and recruited college students and adults from northern, central, and southern Taiwan to participate in a questionnaire survey. The questionnaire comprised a 41-item Inventory for Individual-Oriented View of Self and a 44-item Inventory for Social-Oriented View of Self. Factor analysis was performed to extract four factors each from the two cultural values. The four factors of the individual-oriented view of self were independence, self-determination, competition, and consistency, whereas those of the social-oriented view of self were contextual self, interpersonal relatedness, self-cultivation, and social sensitivity. Numerous studies have reported that the difference between individual and social orientation is also reflected in diverse aspects, such as achievement motivation and self-actualization—two concepts relevant to this study, which are introduced in the following sections.

Easterners and Westerners have distinct behavioral performance and cognitive processes [20–23]. Iyengar and Lepper [20] conducted empirical research to explore the achievement motivation of children from culturally diverse backgrounds. They discovered that the achievement motivation of white children is closely related to self-determination. During the experiment, white children exhibited high motivation only in self-determination situations, whereas Asian children exhibited high motivation and superior behavior in both self-determination and in-group determination situations. Asian children demonstrated particularly high

motivation in in-group determination situations. Accordingly, Asian children are motivated by self-determination and social expectations to pursue achievements. Chen [23] reported that ethnic Chinese people and Westerners differ in achievement motivation, attribution of success and failure, and responses to goal conflicts. People with a high individual-oriented achievement motivation attribute success to internal factors and failure to external factors, and tend to engage in direct communication. People with a high social-oriented achievement motivation attribute success to external factors (no attribution of success when no bystanders are present) and failure to internal factors, and tend to engage in indirect communication when faced with conflicts. In sum, the establishment of achievement motivation is fundamentally different between Westerners and ethnic Chinese people from childhood. In addition, the attribution of success and failure differs between Westerners and ethnic Chinese people. Fundamental attribution error in Western psychology may not apply to ethnic Chinese people because Chinese society expects modesty from individuals. Ethnic Chinese people and Westerners differ considerably in terms of self-actualization and achievement motivation. Yang and Lu [6] combined qualitative and quantitative methods to study the characteristics of eastern and western self-actualizers, and recruited college students, graduate students, and adults from Taiwan and China as participants. They suggested that individual-oriented self-actualizers stand out from the crowd and become independent individuals, whereas social-oriented self-actualizers value their obligations and values regarding relationships with family, groups, country, and society, in addition to expressing their individuality.

## Development of bicultural self

Yang [10] recruited Taiwanese college students as participants and observed that the values of submission to authority, conservation and endurance, and male dominance have been gradually replaced by the value of egalitarianism and open-mindedness. Male dominance has been gradually replaced by gender equality, but filial piety and ancestral worship as well as fatalism and defensiveness still coexist with optimism and assertiveness, independence and self-reliance, and respect for emotions despite changes in the social environment. Said empirical study highlighted that not all traditional Chinese values are replaced by modern values and that the two coexist. Hwang [24] selected a sample of college students and asked them to describe their own values and beliefs as well as those of their parents and older people, thereby exploring generational differences in values. The results indicated that in terms of crucial and core values, the college students did not differ much from their parents. Accordingly, traditional and modern cultural values can coexist and even merge into a new value system, despite generational changes. Lu [25] conducted a focus group with 25 Taiwanese people aged 22–60 years. After compiling various interview data, Lu [25] found that self-construal in modern Taiwanese society is neither completely interdependent (traditional values) nor completely independent (modern and Western values); instead, people employ a composite self-construal process in which the two coexist. According to the aforementioned study results, Taiwanese people in contemporary society endorse both the individual- and social-oriented views of self.

Yang [26] proposed four steps in the evolution or coordination process of the psychology and behavior of individual and social orientation: coexistence→combination→fusion→integration. Through the aforementioned continuous and complex process, ethnic Chinese people integrate the psychology and behavior derived from the social- and individual-oriented views of self with the objective of forming a well-integrated bicultural self. Yang [26] asserted that to form such an ideal self, individuals must continually strengthen their ego strength, ego competence, and ego resiliency during the development of the self. Individuals derived from the manifestation of the bicultural self in thoughts, emotions,

desires, motives, and goals as well as adopt rational methods to eliminate these contradictions and resolve relevant conflicts.

However, according to our observations, college students struggle to form an integrated bicultural identity without appropriate guidance from teachers, peers, and the external environment because contradictory cultural values are conducive to confusion. Without proper guidance and sufficient self-exploration, college students may stay at unintegrated self with low individual- and low social-oriented views. They might lack motivation and enthusiasm for their work, even loss of meaning, goals, and hope. College graduates with unintegrated self-entering the workforce are likely to experience contradictions and conflicts during the integration of two cultural values. To exacerbate matters, they may not know how to approach the situation, lose the meaning and goal of life, and eventually develop demoralization. Additional attention should be paid to this vulnerable group, and they should be provided with relevant support.

## Overview of the present study

Nowadays, college students usually feel confused, loss motivation and have demoralization about their life. Taiwanese college students are simultaneously subject to two cultural values coexist within each individual; therefore, they may have conflicts and contradictions during the development of self-identity. The goals of this study focused on the cultural factors contributing to college students' demoralization to explore the effect of two cultural value systems on demoralization. Three research questions were included in this study: (a) what is the students' demoralization status, (b) what is the student' individual- and social-oriented views of self? The predominate research question driving this study is, what is the relationship between students' individual- and social-oriented views of self and their demoralization among college student in Taiwan?

## Methods

### Participants

Taiwanese students ($N$ = 356) from various universities participated in this study. They completed the questionnaire with an online questionnaire. Table 1 lists the distribution of the sample, which excluding 110 incomplete questionnaires and excluding 86 questionnaires to balance variables such as sex and school region. The study was approved by and following the guideline of the local ethics committee (Research Ethics Committee at the National Taiwan University, IRB no. 202104ES027). All participants read and agreed online informed consent.

### Instruments

All data are available in S1 Appendix.

**Mandarin version of Demoralization scale (DS-M-25).** The Mandarin version of Demoralization scale (DS-M-25) was guided by the demoralization scale for cancer patients

**Table 1. Distribution of the sample.**

| School region | Freshman (men/women) | Sophomore (men/women) | Junior (men/women) | Senior (men/women) | Regional total |
|---|---|---|---|---|---|
| Northern Taiwan | 20/15 | 16/22 | 9/15 | 8/13 | 53/65 |
| Central Taiwan | 20/16 | 13/13 | 25/14 | 11/21 | 69/64 |
| Southern Taiwan | 10/21 | 10/14 | 12/19 | 9/10 | 41/64 |
| Total | 50/52 | 39/49 | 46/48 | 28/44 | 163/193 |

[16] and its translated Mandarin version as DS-M-25 scale [2, 27]. In the DS-M-25 scale, the measurement (of demoralization) was including five factors comprising the following dimensions: life meaning; loneliness and helplessness; self-assurance; bravery and perseverance; and emotional distress. Five items were generated for each dimension, since the scale was designed for adolescents, negative-worded items were minimized to reduce the cognitive burden of participants. The 25 items using a 5-point Likert scale (1: *strongly disagree* to 5: *strongly agree*) to indicate the extent to which they agreed (or disagreed) with the items. Specifically, it shows good internal consistency and test–retest reliability for both adolescent and young adults [17]. The instrument was similar to our prior study, which was including detailed validity of DS-M scale [17, 27].

**Individual- and Social-oriented Self scale (ISS).**   The Individual- and Social-oriented Self scale (ISS) was developed with dual 4-factor structure of the ISS subscales encompassed the key dimensions of the initial theoretical frameworks and broadly covers well-documented cultural differences in the self, including independence versus interdependence, cross-situational consistency versus context sensitivity, and other- versus self-focus [19, 28]. The 40-item ISS scale was a conglomeration of the individual- versus social-oriented self-dichotomy [28]. The measurement of individual-oriented self (ISS-I) was including four factors comprising the following dimensions: independence; self-determination; competition; and consistency. In addition, the measurement of social-oriented self (ISS-S) was including four factors comprising the following dimensions: contextual self; interpersonal relatedness; self-cultivation; and social sensitivity. Five items were generated for each dimension using a 5-point Likert scale (1: strongly disagree to 5: strongly agree) to indicate the extent to which they agreed (or disagreed) with the items. According to previous studies, the ISS scale has revealed good reliability and validity [19, 28–31].

## Data analysis

For each scale, the overall score was based on all dimensions; in addition, sub-scores were formed with 5-item in each dimension. SPSS package version 21.0 was used for data analysis, and three main analysis methods were adopted. First, one-way ANOVAs were used to examine the differences among dimensions in each scale. Then, we used the mean of the overall ISS-I/ ISS-S score as the cut-off point, and a 2×2 factorial design was allowed us to examine the effects of ISS-I/ISS-S (each with high/low levels) on their demoralization. Therefore, all participants were divided into four groups, namely bicultural self (high ISS-I × high ISS-S), individual-oriented self (high ISS-I × low ISS-S), social-oriented self (low ISS-I × high ISS-S), and unintegrated self (low ISS-I × low ISS-S). Two-way ANOVAs were to explore the differences in the demoralization scores of college students with distinct individual- or social-oriented views of self. Subsequently, Scheffe's post hoc test was performed. Third, the multiple regression model was used to explore the association between demoralization score and the measurement of individual-oriented self (ISS-I)/social-oriented self (ISS-S) among 356 participants. The $\alpha$ value was at the level of adjusted $p$ values $< 0.05$. for the analysis.

## Results

## College students' demoralization, individual- and social-oriented view of self

Table 2 presents the demoralization, ISS-I and ISS-S scores in the four groups. In the demoralization scores, "emotional distress" was significant higher than other dimensions, indicating that they face uncertainty concerning their future career during their current life stage, which

Table 2. College students' demoralization and individual/social-oriented view of self.

|  | Bicultural self (n = 99) | Individual-oriented (n = 73) | Social-oriented (n = 63) | Unintegrated self (n = 121) | All participants (n = 356) |
|---|---|---|---|---|---|
| *Demoralization* |  |  |  |  |  |
| Meaning of life | 2.21±0.65 | 2.56±0.72 | 2.58±0.80 | 2.86±0.68 | 2.57±0.74 |
| Loneliness and helplessness | 2.51±0.77 | 2.67±0.77 | 2.46±0.86 | 2.58±0.77 | 2.46±0.77 |
| Self-affirmation | 2.07±0.62 | 2.31±0.74 | 2.60±0.82 | 2.79±0.72 | 2.56±0.79 |
| Courage and tenacity | 1.84±0.49 | 2.15±0.51 | 2.40±0.62 | 2.60±0.54 | 2.26±0.62 |
| Emotional distress | 2.65±0.63 | 2.76±0.64 | 3.14±0.69 | 3.12±0.64 | 2.92±0.68 |
| *Overall score* | 2.26±0.50 | 2.49±0.53 | 2.64±0.62 | 2.79±0.54 | 2.55±0.58 |
| *Individual-oriented view of self* |  |  |  |  |  |
| Independence | 4.42±0.34 | 4.22±0.39 | 4.09±0.40 | 3.83±0.51 | 4.12±0.48 |
| Autonomy | 3.58±0.57 | 3.42±0.61 | 2.52±0.66 | 2.66±0.52 | 3.05±0.73 |
| Competition | 3.47±0.63 | 3.14±0.61 | 2.62±0.53 | 2.57±0.52 | 2.95±0.69 |
| Consistency | 3.44±0.71 | 3.18±0.63 | 2.45±0.70 | 2.48±0.56 | 2.88±0.78 |
| *Overall ISS-I score* | 3.72±0.36 | 3.49±0.21 | 2.92±0.20 | 2.88±0.23 | 3.25±0.46 |
| *Social-oriented view of self* |  |  |  |  |  |
| Situational self | 4.32±0.38 | 3.88±0.43 | 4.27±0.39 | 3.82±0.48 | 4.05±0.48 |
| Interpersonal connection | 3.86±0.51 | 2.93±0.59 | 3.66±0.52 | 2.90±0.49 | 3.31±0.68 |
| Self-cultivation | 4.06±0.44 | 3.32±0.46 | 3.79±0.36 | 3.17±0.40 | 3.56±0.56 |
| Social sensitivity | 3.00±0.35 | 2.45±0.53 | 2.96±0.40 | 2.50±0.42 | 2.71±0.49 |
| *Overall ISS-S score* | 3.81±0.28 | 3.15±0.24 | 3.67±0.18 | 3.10±0.24 | 3.41±0.40 |

*Note.* M, mean; SD, standard deviation; ISS-I, individual-oriented self; ISS-S, social-oriented self.

may result in dysphoria and a sense of lack of control. Also, the "bravery and perseverance" was significant lower than other dimensions, suggesting that they have more opportunities for self-development during their time in college; some students even moved out of their parents' home for college, which enables them to learn and take responsibility for themselves.

In the ISS scale, college students scored the highest on independence—pertaining to the individual-oriented view of self—followed by contextual self, self-cultivation, and interpersonal relatedness—pertaining to the social-oriented view of self. Self-determination, with the fifth highest score, is associated with the individual-oriented view of self. There was no significant difference between independence and contextual self. Taiwanese college students live in an integrated bicultural environment, rather than in a context where individual orientation overpowers social orientation; this concurs with the bicultural self perspective [10, 24, 25, 32–34]. Although academia is dominated by personal orientation and emphasizes the value of independent thinking and critical thinking [35], social-oriented cultural values such as relationship orientation, interpersonal affection, and face [36] still substantially influence college students' interaction.

## Differences in the influence on demoralization level between individual- and social-oriented views of self

The mean of the overall ISS-I/ISS-S score was as the cut-off point, and a 2×2 factorial design was allowed us to examine the effects of ISS-I/ISS-S on their demoralization. Four groups with bicultural self (high ISS-I × high ISS-S), individual-oriented self (high ISS-I × low ISS-S), social-oriented self (low ISS-I × high ISS-S), and unintegrated self (low ISS-I × low ISS-S)

**Table 3. Demoralization overall score among four groups.**

| Grouping | Demoralization overall score (M ± SD) | F-value | Scheffe's post-hoc test |
|---|---|---|---|
| Bicultural self (n = 99) | 2.26±0.50 | 18.44*** | U > B; |
| Individual-oriented self (n = 73) | 2.49±0.53 | | U > I; |
| Social-oriented self (n = 63) | 2.64±0.62 | | S > B; |
| Unintegrated self (n = 121) | 2.79±0.54 | | I > B |

*Note.* N, number; M, mean; SD, standard deviation; B, Bicultural self; I: Individual-oriented self; S: Social-oriented self; U, Unintegrated self;

***$p < .001$.

comprised 99, 73, 63, and 121 college students. Table 3 presents the results of demoralization scores yielded by these four groups.

The demoralization overall score of bicultural self group was significant lower than other groups, indicating college students with an integrated bicultural self help them to reduce the occurrence of demoralization. Accordingly, college students with a bicultural view of self were more adaptive than their counterparts with either an individual- or social-oriented view of self. A higher level of demoralization was observed in the unintegrated self group than in the individual-oriented group possibly because college students develop their ideal self on the basis of individual orientation and develop their ought self according to social orientation. The ideal self facilitates the establishment of career goals and prevents occurrence of demoralization.

In sum, the current results suggest that college students' identification with cultural value influences the occurrence of demoralization. College students who identified with bicultural values were less demoralized than those who identified with either or neither of the cultural values, which implies that college students with an integrated bicultural view of self are more adaptive than their counterparts who endorse either the individual- or social-oriented view of self.

## Correlations of demoralization with individual- and social-oriented view of self

Table 4 presents the associations between demoralization score and the measurement of individual-oriented self (ISS-I)/social-oriented self (ISS-S). For ISS-I, moderate to weak negative correlations were observed between demoralization scores and individual-oriented view of self. Competition was significant positively correlated with loneliness and helplessness, suggesting that competition involves affirming self-worth through interpersonal competition and personal achievement. A previous study had explored Taiwanese college students' suppression of emotional expression in interpersonal context and discovered that they refrain from expressing positive emotions to avoid excessive bragging and attracting jealousy and resentment [37]. Undergraduate education in Taiwan encourages cooperation and emphasizes management of interpersonal relationships. College students who are excessively competitive with their classmates easily become socially excluded, which makes them feel frustrated, lonely, and helpless in interpersonal relationships. Competition is a cultural value that is prevalent in the west, but it may not be endorsed by Taiwanese college students because it does not facilitate adaptability and development of self-worth. College students who overemphasize peer competition are unlikely to obtain a positive meaning of life and likely to experience loneliness.

For ISS-S, moderate to weak negative correlations were observed between demoralization scores and social-oriented view of self. Social sensitivity was significant positively correlated

**Table 4. Association between demoralization score and individual-oriented self (ISS-I)/social-oriented self (ISS-S) among 356 participants.**

| | Meaning of life | | Loneliness & helplessness | | Self-assurance | | Courage and tenacity | | Emotional distress | | Demoralization Overall score | |
|---|---|---|---|---|---|---|---|---|---|---|---|---|
| | β | p | β | p | β | p | β | p | β | p | β | p |
| *Individual-oriented view of self* | | | | | | | | | | | | |
| **Independence** | -.298 | <.001 | -.330 | <.001 | -.199 | <.001 | -.447 | <.001 | -.088 | .071 | -.334 | <.001 |
| **Self-determination** | -.129 | .015 | -.352 | <.001 | .009 | .874 | -.280 | <.001 | -.338 | <.001 | -.263 | <.001 |
| **Competition** | .111 | .026 | .152 | .001 | .326 | <.001 | .076 | .071 | .150 | .002 | .208 | <.001 |
| **Consistency** | -.255 | <.001 | -.190 | <.001 | -.206 | <.001 | -.193 | <.001 | -.302 | <.001 | -.284 | <.001 |
| *Overall score* | -.312 | <.001 | -.478 | <.001 | -.014 | .808 | -.495 | <.001 | -.469 | <.001 | -.426 | <.001 |
| *Social-oriented view of self* | | | | | | | | | | | | |
| **Contextual self** | -.169 | .002 | -.229 | <.001 | -.144 | .012 | -.300 | <.001 | -.001 | .984 | -.207 | <.001 |
| **Interpersonal relatedness** | -.264 | <.001 | -.086 | .136 | -.162 | .010 | -.006 | .919 | -.137 | .027 | -.168 | .004 |
| **Self-cultivation** | -.147 | .019 | -.309 | <.001 | -.039 | .549 | -.396 | <.001 | -.207 | .002 | -.263 | <.001 |
| **Social sensitivity** | .205 | <.001 | .310 | <.001 | .254 | <.001 | .222 | <.001 | .236 | <.001 | .306 | <.001 |
| *Overall score* | -.153 | .005 | -.036 | .489 | -.072 | .227 | -.139 | .005 | .095 | .079 | -.075 | .154 |

with loneliness and helplessness, self-assurance, and emotional distress, suggesting that social sensitivity means to value interpersonal harmony and pay special attention to the evaluation of others. A previous study had showed that college students' fear of negative evaluation was positively correlated with depression, negatively correlated with social self-efficacy, and positively correlated with shyness [38]. College students who are afraid of negative evaluation tend to feel nervous and are unlikely to affirm themselves. Also, a weak and positive correlation was observed between social sensitivity and loneliness and helplessness, self-assurance, and emotional distress, suggesting that those students without sufficient social skills tend to feel lonely and helpless. A social-oriented view of self helped reduce demoralization, but social sensitivity strengthened these three dimensions of demoralization. As explained in the previous chapter, well-managed interpersonal relationships that are based on the social-oriented self can prevent demoralization, whereas excessive concern over others' opinions aggravates demoralization.

In summary, college students with a well-integrated bicultural self were significantly less demoralized than those with an unintegrated self. Most dimensions of demoralization in college students were significant negatively correlated with individual- and social-oriented views of self. Accordingly, college students' individual- and social-oriented views of self may prevent the occurrence of demoralization. However, competition was positively correlated with loneliness and helplessness, whereas social sensitivity was weakly positively correlated with loneliness and helplessness, self-assurance, and emotional distress, suggesting that two dimensions (i.e., competition and social sensitivity) had little effect on college students' demoralization.

## Discussion

### Traditional Chinese cultural values coexist with modern values rather than being replaced

College students had significant higher scores of the emotional distress than on other demoralization dimensions. Accordingly, college students' demoralization mostly manifests in uneasiness and anxiety. The factors of individual- and social- oriented views of self with the highest to lowest influence on the participants are as follows: independence, contextual self, self-cultivation, interpersonal relatedness, self-determination, competition, consistency, and social sensitivity. These top ranking factors comprise a mixture of elements from both individual- and

social-oriented views of self, suggesting that Taiwanese college students might have bicultural self to influence their thoughts and interpersonal relationships. Of all factors of self-view in college students, competition and social sensitivity were the only factors that aggravate college students' demoralization.

Numerous studies have supported that modern-day ethnic Chinese people live in a bicultural environment in which they develop a bicultural self that integrates individual- and social-oriented views of self [18, 24, 25, 32–34]. According to the current results, three of the top five self-view factors ranked by college students are social-oriented factors, which indicates that traditional Chinese cultural values have been subtly incorporated into modern thought, rather than disappearing over time. Our findings are consistent with the perspective of Yang [26] that the formation of the Chinese bicultural self is a continuous process. The binary opposite system of self enables Taiwanese college students who are subject to Chinese culture to adjust themselves constantly in response to temporal and spatial changes over time.

The top four cultural values endorsed by Taiwanese college students were independence, contextual self, self-cultivation, and interpersonal relatedness, whereas the bottom three were competition, consistency, and social sensitivity. The cultural values of Taiwanese college students are manifested in independent thinking, emphasis on interpersonal cooperation, and self-expression adjustment according to different situations, occasions, and role-relationships. The self-view of modern-day Taiwanese college students includes individual-oriented dependent thinking while retaining the social-oriented interdependent and ensembled self because they still wish to maintain a harmonious relationship with others. These aforementioned finding concurs with the concept of composite self proposed by Lu [25]. Therefore, college students still incorporate the value of interdependence into their everyday practices while pursuing independence, such an integrated composite self in turn affects individual adaptation to life changes.

## College students' cultural value identification and integration affects their demoralization level

According to the demoralization, the bicultural self, individual-oriented self, and social-oriented self groups had lower scores than the unintegrated self group. This indicates that college students' development of any type of self-view can help reduce the occurrence of demoralization. College students with an integrated bicultural self were less demoralized than were their counterparts who identified with either or neither of the two cultural value systems. Demoralization results from individuals' difficulties in identifying goals and the meaning of life [5, 12, 39]. Cultural values facilitate individuals' development of meaning in life [6–8], which in turn affects the occurrence of demoralization. The current study discovered that both individual- and social-oriented cultural values were significantly correlated with demoralization. The bicultural self group had a significant lower demoralization than the unintegrated self group. Our result is consistent with previous studies on bicultural self, which have advocated that a bicultural composite self positively affects the adaptability of individuals [40–43].

From the perspective of cultural psychology, self-construal reflects people's thoughts, emotions, and actions as well as how they view the world, themselves, and the external world, including their relationships and other people and things, all of which affect how they think and act [44]. Two considerably different cultural values coexist in Taiwan's social environment, including individual- and social-oriented cultural values [9]. The lack of compatibility and complementarity between these two cultural value systems may result in cultural conflicts during the self-construal of college students, which in turn affects their demoralization level.

People who fail to integrate said two cultural systems and develop a composite self-construal—similar to the effect of failing to integrate internal self-discrepancy—are likely to experience anxiety, disappointment, sadness, and depression, eventually losing their life goals and developing demoralization. Therefore, the social environment in which people live determines the type of people they wish to become [6]; these cultural values shape people's view of self and meaning of life [8], eventually affecting the occurrence of demoralization.

## College students' demoralization is associated with cultural compatibility

The ranking of the influence of individual- and social-oriented cultural values on college students, from highest to lowest, is as follows: independence, contextual self, self-cultivation, interpersonal relatedness, self-determination, competition, consistency, and social sensitivity. Of these cultural values, competition was positively correlated with loneliness and helplessness, and social sensitivity was positively correlated with loneliness and helplessness, emotional distress, and self-assurance. College students identified the least with competition and social sensitivity, both of which contributed to the occurrence of demoralization. Thus, the consistency between college students' view of self and the culture they are exposed to directly affected the occurrence of demoralization, which concurs with related literature advocating that bicultural adaptation difficulties can lead to demoralization.

Yeh, Bedford [45] proposed the concept of relating autonomy in the dual autonomy model of Chinese people, which represents the volitional self-adjustment of Chinese people to maintain a harmonious and dependent connection with significant others because they value interpersonal harmony. Although Taiwanese college students pursue interpersonal harmony, this study revealed that they still prioritize independence as the most crucial value they express in private contexts. That is, Taiwanese college students may still maintain certain levels of self-determination, which they hide from others, in the pursuit of interpersonal harmony. Such a phenomenon resembles self-effacing behavior in ethnic Chinese people, which is in fact tactical self-enhancement in disguise. Ethnic Chinese people do not truly believe that they are inadequate or inferior, and they only self-efface to achieve interpersonal harmony and conform to social expectations [46, 47]. Whether the current findings are the result of the influence of doctrine of the mean or self-cultivation—emphasized in the Chinese cultural system—warrants further investigation.

This study is limited by a lack of data to distinguish the developmental changes of the individual- and social-oriented views of self. Moreover, the small sample size of this study may not have enough power to demonstrate the influences of different cultural values. Therefore, it warrants further investigations with larger samples and different experimental designs to validate our findings.

## Conclusion

Despite increased screening for depression in colleges, demoralization symptoms in college students are often either missed or dismissed. Our results suggest that it is important to help students to develop a bicultural self to cope with changes in time. Furthermore, in college, teachers should be able to identify and recognize the demoralization symptoms, is an important goal for future research.

## Supporting information

**S1 Appendix.**
(XLSX)

## Author Contributions

**Conceptualization:** Chuan-Yung Huang.

**Data curation:** Shun-Hao Hu, Li-Ying Fan.

**Project administration:** Chuan-Yung Huang.

**Supervision:** Chuan-Yung Huang.

**Validation:** Li-Ying Fan.

**Writing – original draft:** Chuan-Yung Huang, Shun-Hao Hu, Li-Ying Fan.

**Writing – review & editing:** Li-Ying Fan.

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
