## [Decision Letter · Decision Letter 0]

2 Dec 2021

PONE-D-21-20917

Relationship between Demoralization of the College Student with their Individual- and Social-Oriented Self

PLOS ONE

Dear Dr. Fan,

Thank you for submitting your manuscript to PLOS ONE. After careful consideration, we feel that it has merit but does not fully meet PLOS ONE’s publication criteria as it currently stands. Therefore, we invite you to submit a revised version of the manuscript that addresses the points raised during the review process.

I went over the manuscript and agreed with the two reviewers to recommend major revision for the following reasons:

• The introduction needs to be revised. For example, there is quite more words on the description of depression.

• The theoretical and conceptual value of this paper is unclear.

• It is not very clear how this paper builds from the extant literature and fills the important gap in this field.

• The data collection part is missing - how to collect the data for the study and when.

• The conclusion section should consider the limitations of the research, causes and solutions, etc.

• End the discussion elucidating the important contribution to the literature.

• Reference needs to be improved, such as formatting needs to match, and selection of more literature from regions other than Taiwan may be more representative.

We hope that these recommendations, when considered, will improve the quality of manuscript notch higher. We therefore request you to revise the paper following my and reviewer's recommendations.

Thanks so much.

We look forward to receiving your revised manuscript.

Kind regards,

Fraide Agustin Ganotice, PhD

Academic Editor

PLOS ONE

2. Please change "female” or "male" to "woman” or "man" as appropriate, when used as a noun (see for instance https://apastyle.apa.org/style-grammar-guidelines/bias-free-language/gender).

Reviewers' comments:

Reviewer's Responses to Questions

**Comments to the Author**

1. Is the manuscript technically sound, and do the data support the conclusions?

Reviewer #1: Yes

Reviewer #2: Partly

2. Has the statistical analysis been performed appropriately and rigorously? 

Reviewer #1: N/A

Reviewer #2: I Don't Know

3. Have the authors made all data underlying the findings in their manuscript fully available?

Reviewer #1: Yes

Reviewer #2: Yes

4. Is the manuscript presented in an intelligible fashion and written in standard English?

Reviewer #1: Yes

Reviewer #2: Yes

5. Review Comments to the Author

Reviewer #1: The manuscript is interesting with significant volume of informations. The research instruments are adequate for study organisation and qualitative data. The results and discussions should appear to be related with their individual social oriented self. The conclusions are relevant for the topic. I consider that the authors do a good job and the manuscript and the subject matter is very interesting for the journal.

1. It seems that regression analysis is necessary to analyze the purpose of the study. Correlation analysis alone is problematic in interpreting the relationship between variables.

2. Previous studies are mainly Taiwanese journals. If you present a foreign journal together, it will be a better thesis.

3. The bibliography presentation in the text must be matched. Please match either year or number according to the academic form.

Reviewer #2: Background

1. Other than the meaning of life, the authors should also elaborate more on how "low individual- and low social-oriented self" would influence/inform other components of demoralization (e.g, self-affirmation, courage and tenacity, emotional distress).

Methods/Materials

2. The authors should mention how the total scores of demoralization, individual- and social-oriented self were computed.

3. The authors should indicate if there are any previous studies supporting the validity of the two scales.

Data Analysis

4. Is there any reason(s) why the mean scores of the participants' individual and social orientation were used as the cutoff point? Given that the mean scores of "Independence" and "Situational self" were relatively high (over 4), low individual orientation/low social orientation groups may contain students who were high in independence and situational self.

5. The authors should mention the statistical procedure used to rank the individual- and social-oriented cultural values.

Results

6. Table 2 should report the descriptive statistics of the total scores of demoralization, individual- and social-oriented self. Similarly, table 4 should report the correlations between the total scores of demoralization, individual- and social-oriented self.

7. Given that the study hypothesized that "low individual- and low social-oriented self" would inform the development of demoralization, it is not clear that the findings of "competition" and "social sensitivity" contradicted the hypothesis.

8. Were any variables considered as possible co-variates?

Discussion

9. A discussion of limitations of the study should be included.

6. PLOS authors have the option to publish the peer review history of their article (what does this mean?). If published, this will include your full peer review and any attached files.

Reviewer #1: No

Reviewer #2: No

---

## [Author Response · Author response to Decision Letter 0]

16 Feb 2022

Dear Editor:

Thank you for giving us the opportunity to revise and resubmit our manuscript entitled “Relationship between Demoralization of the College Student with their Individual- and Social-Oriented Self” to PLOS ONE. The submitted manuscript has been approved by all the authors. We would like to thank you and the reviewers for the valuable comments.

We have revised this manuscript according to the comments of reviewers. In addition, we have provided our detailed responses to the comments following this cover letter. The original manuscript number was PONE-D-21-20917. Uploaded, please kindly find the revised manuscript showing the changes in yellow highlights [PONE-D-21-20917 (Revised Manuscript with Track Changes)] and a clean copy of the revised manuscript [PONE-D-21-20917 (manuscript)]. Our point-by-point response to each comment from the reviewers [PONE-D-21-20917 (Response to Reviewers)] and this cover letter [PONE-D-21-20917 (cover letter)] are uploaded. 

The present study is the first study the extent of the relationship between attitude towards demoralization, individual- and social-oriented views of self in college students. We found that college students with a bicultural view of self-demonstrated significantly lower levels of demoralization than other groups. Moreover, the five components of demoralization in college students were mostly significantly and negatively correlated with individual- and social-oriented views of self, indicating that college students’ bicultural views of self may contribute to or prevent demoralization.

We appreciate the helpful comments from the reviewers and the editor to strengthen the ultimate yield of this paper. We hope that this revised revision now meets the high standards of PLOS ONE and can be accepted for publication.

Sincerely yours,

Chuan-Yung Huang, Ph.D.

Li-Ying Fan, Ph.D.

---

## [Decision Letter · Decision Letter 1]

23 Jun 2022

Relationship between Demoralization of the College Student with their Individual- and Social-Oriented Self

PONE-D-21-20917R1

Dear Dr. Fan,

We’re pleased to inform you that your manuscript has been judged scientifically suitable for publication and will be formally accepted for publication once it meets all outstanding technical requirements.

Kind regards,

Fraide Agustin Ganotice, PhD

Academic Editor

PLOS ONE

Additional Editor Comments (optional):

Dear Authors,

Thank you for submitting your manuscript titled "Relationship between Demoralization of the College Student with their Individual- and Social-Oriented Self” to PLOS ONE. Two reviewers provided their take on the revised version and suggested acceptance. I am here to make a recommendation of acceptance too.

Before the paper’s publication, I hope you can consider the comments and suggestions given by the reviewers to improve your final manuscript.

Thanks so much.

Respectfully yours,

Fred Ganotice

Reviewers' comments:

Reviewer's Responses to Questions

**Comments to the Author**

1. If the authors have adequately addressed your comments raised in a previous round of review and you feel that this manuscript is now acceptable for publication, you may indicate that here to bypass the “Comments to the Author” section, enter your conflict of interest statement in the “Confidential to Editor” section, and submit your "Accept" recommendation.

Reviewer #2: All comments have been addressed

Reviewer #3: All comments have been addressed

2. Is the manuscript technically sound, and do the data support the conclusions?

Reviewer #2: Yes

Reviewer #3: Yes

3. Has the statistical analysis been performed appropriately and rigorously? 

Reviewer #2: I Don't Know

Reviewer #3: Yes

4. Have the authors made all data underlying the findings in their manuscript fully available?

Reviewer #2: Yes

Reviewer #3: Yes

5. Is the manuscript presented in an intelligible fashion and written in standard English?

Reviewer #2: Yes

Reviewer #3: Yes

6. Review Comments to the Author

Reviewer #2: The authors have been very responsive in responding to my comments, and my only remaining suggestions are minor, and I leave them to the discretion of the editor and authors.

Thank you very much for the opportunity to review this important research.

1. Consider adding more literature review and previous research findings on “unintegrated-self” (low in both individual- and social-oriented self) in the Introduction, as this is the most vulnerable group to demoralisation suggested in this research.

Reviewer #3: The research is oriented to the study of a relevant topic for cultural psychology. According to the structure of the manuscript: (1) The research problem is clearly described, making a coherent construction of the object of study, (2) The references used are relevant, (3) A methodological strategy adequately supported by the theory is proposed to achieve the stated objectives, (4) The presentation of the results is coherent and clear, (5) The discussion and conclusion are based on the results and adequate references are included to support the findings.

7. PLOS authors have the option to publish the peer review history of their article (what does this mean?). If published, this will include your full peer review and any attached files.

Reviewer #2: No

Reviewer #3: **Yes: **Jorge Vergara-Morales

---

## [Editor Report · Acceptance letter]

27 Jun 2022

PONE-D-21-20917R1 

Relationship between Demoralization of the College Student with their Individual- and Social-Oriented Self 

Dear Dr. Fan:

I'm pleased to inform you that your manuscript has been deemed suitable for publication in PLOS ONE. Congratulations! Your manuscript is now with our production department. 

Kind regards, 

on behalf of

Dr. Fraide Agustin Ganotice 

Academic Editor

PLOS ONE